# Gen𝒳D: Generating any 3D and 4D Scenes

**Yuyang Zhao♣, Chung-Ching Lin♠, Kevin Lin♠, Zhiwen Yan♣, Linjie Li♠,**
**Zhengyuan Yang♠, Jianfeng Wang♠, Gim Hee Lee♣, Lijuan Wang♠**
♣ National University of Singapore, ♠ Microsoft Corporation
https://gen-x-d.github.io

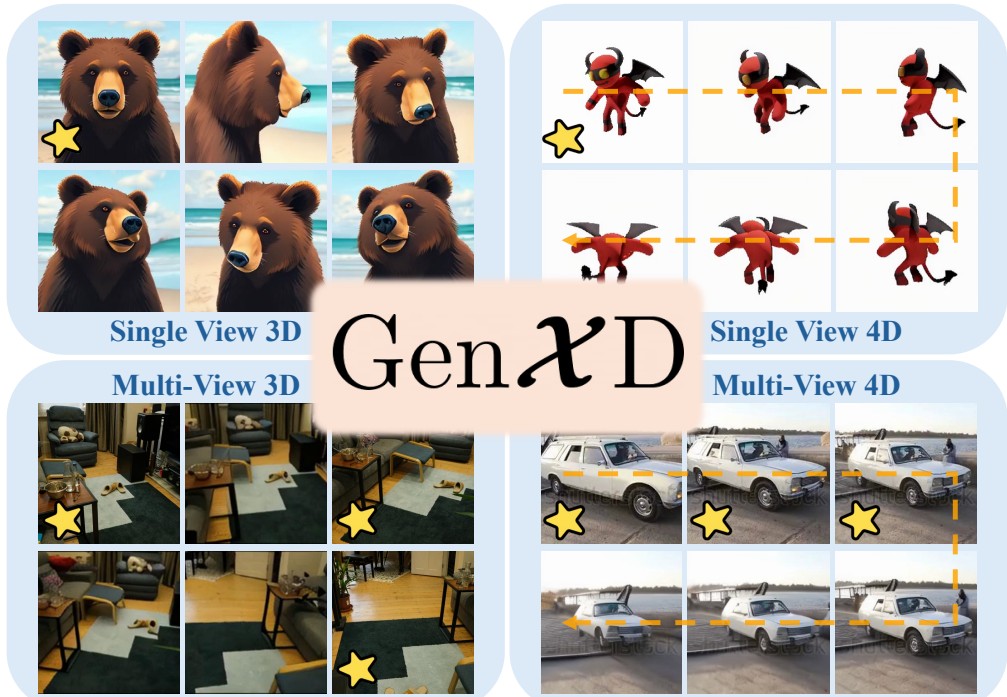

Figure 1: Gen𝒳D is a unified model for high-quality 3D and 4D generation from any number of condition images. By controlling the motion strength and condition masks, Gen𝒳D can support various application without any modification. The condition images are shown with **star icon** and the time dimension is illustrated with **dash line**.

## ABSTRACT

Recent developments in 2D visual generation have been remarkably successful. However, 3D and 4D generation remain challenging in real-world applications due to the lack of large-scale 4D data and effective model design. In this paper, we propose to jointly investigate general 3D and 4D generation by leveraging camera and object movements commonly observed in daily life. Due to the lack of real-world 4D data in the community, we first propose a data curation pipeline to obtain camera poses and object motion strength from videos. Based on this pipeline, we introduce a large-scale real-world 4D scene dataset: **CamVid-30K**. By leveraging all the 3D and 4D data, we develop our framework, **Gen𝒳D**, which allows us to produce any 3D or 4D scene. We propose multiview-temporal modules, which disentangle camera and object movements, to seamlessly learn from both 3D and 4D data. Additionally, Gen𝒳D employs masked latent conditions to support a variety of conditioning views. Gen𝒳D can generate videos that follow the camera trajectory as well as consistent 3D views that can be lifted into 3D representations. We perform extensive evaluations across various real-world and synthetic datasets, demonstrating Gen𝒳D's effectiveness and versatility compared to previous methods in 3D and 4D generation.

## 1  INTRODUCTION

Generating 2D visual content has achieved remarkable success with diffusion (Rombach et al., 2022; Betker et al., 2023; Esser et al., 2024; Blattmann et al., 2023) and autoregressive modeling (Tian et al., 2024; Sun et al., 2024; Kondratyuk et al., 2023; Luo et al., 2024), which have already been used in real-world applications, benefiting society. In addition to 2D generation, 3D content generation is also of vital importance, with applications in video games, visual effects, and wearable mixed reality devices. However, due to the complexity of 3D modeling and the limitations of 3D data, 3D content generation is still far from satisfactory and is attracting more attention. In this paper, we focus on the unified generation of 3D and 4D content. Specifically, static 3D content involves only spatial view changes, referred to as *3D generation* in this paper. In contrast, dynamic 3D content includes movable objects within the scene, requiring the modeling of both spatial view and dynamic (temporal) changes, which we term *4D generation*.

Most previous works (Liu et al., 2023b; Shi et al., 2023; Zhao et al., 2023; Xie et al., 2024; Tang et al., 2024; 2023) focus on 3D and 4D generation using synthetic object data. Synthetic object data are typically meshes, allowing researchers to render images and other 3D information (*e.g.*, normals and depth) from any viewpoint. However, object generation is more beneficial to specialists than to the general public. In contrast, scene-level generation can help everyone enhance their images and videos with richer content. As a result, recent works (Gao et al., 2024; Wu et al., 2024b) have explored general 3D generation (both scene-level and object-level) in a single model, achieving impressive performance. Nonetheless, these works focus solely on static 3D generation, without addressing dynamics. In this paper, we propose a *unified framework for general 3D and 4D generation*, enabling the generation of images from different viewpoints and timesteps with any number of conditioning images (Fig. 1).

The first and foremost challenge in 4D generation is the lack of general 4D data. In this work, we propose **CamVid-30K**, which contains approximately 30K 4D data samples. 4D data require both multi-view spatial information and temporal dynamics, so we turn to video data to obtain the necessary 4D data. Specifically, we need two key attributes from the video: the camera pose for each frame and the presence of movable objects. To achieve this, we first estimate the possible movable objects in the video using a segmentation model and then estimate the camera pose using keypoints in the static parts of the scene. While successful camera pose estimation ensures multiple views, we also need to ensure that moving objects are present in the video, rather than purely static scenes. To address this, we propose an object motion field that leverages aligned depth to estimate true object movement in the 2D view. Based on the object motion field, we filtered out static scenes, resulting in approximately 30K videos with camera poses.

In addition, we propose a unified framework, **Gen$\mathcal{X}$D**, to handle 3D and 4D generation within a single model. While there are similarities between 3D and 4D data in terms of their representation of spatial information, they differ in how they capture temporal information. Therefore, 3D and 4D generation can complement each other through the disentanglement of spatial and temporal information (see Appendix. G for details). To achieve this, we combine both 3D and 4D data during model training. To disentangle the spatial and temporal information, we introduce multiview-temporal modules in Gen$\mathcal{X}$D. In each module, we use $\alpha$-fusing to merge spatial and temporal information for 4D data, while removing temporal information for 3D data. Previous works (Xu et al., 2024; Voleti et al., 2024) typically use a fixed number of conditioning images (*e.g.*, the first image). However, single-image conditioning can be more creative, whereas multi-image conditioning offers greater consistency. As a result, we implement masked latent conditioning in our diffusion model. By masking out the noise in the conditioning images, Gen$\mathcal{X}$D can support any number of input views without modifying the network. With high-quality 4D data and a 4D spatio-temporal generative model, Gen$\mathcal{X}$D achieves significant performance in both 3D and 4D generation using single or multiple input views. Our contributions are summarized as follows:

- We design a data curation pipeline for obtaining high-quality 4D data with movable objects from videos and annotate 30,000 videos with camera poses. This large-scale dataset, termed CamVid-30K, will be made available for public use.

- We propose a 3D-4D joint framework, Gen$\mathcal{X}$D, which supports image-conditioned 3D and 4D generation in various settings (Tab. 1). In Gen$\mathcal{X}$D, the multiview-temporal layer is introduced to disentangle and fuse multi-view and temporal information.

Table 1: Comparison among the settings of previous works.

| Method | 3D Generation | | | | 4D Generation | | | |
|---|---|---|---|---|---|---|---|---|
| | Object | Scene | Single View | Multi-View | Object | Scene | Single View | Multi-View |
| IM-3D | ✓ | ✗ | ✓ | ✗ | ✗ | ✗ | ✗ | ✗ |
| RealmDreamer | ✗ | ✓ | ✓ | ✗ | ✗ | ✗ | ✗ | ✗ |
| ReconFusion | ✓ | ✓ | ✗ | ✓ | ✗ | ✗ | ✗ | ✗ |
| CAT3D | ✓ | ✓ | ✓ | ✓ | ✗ | ✗ | ✗ | ✗ |
| Animate124 | ✗ | ✗ | ✗ | ✗ | ✓ | ✗ | ✓ | ✗ |
| CameraCtrl | ✗ | ✗ | ✗ | ✗ | ✗ | ✓ | ✓ | ✗ |
| SV4D | ✓ | ✗ | ✓ | ✗ | ✓ | ✗ | ✓ | ✓ |
| CamCo | ✗ | ✓ | ✓ | ✗ | ✗ | ✓ | ✓ | ✗ |
| MotionCtrl | ✗ | ✓ | ✓ | ✗ | ✗ | ✓ | ✓ | ✗ |
| **Gen$\mathcal{X}$D (Ours)** | ✓ | ✓ | ✓ | ✓ | ✓ | ✓ | ✓ | ✓ |

- Using the proposed CamVid-30K along with other existing 3D and 4D datasets, Gen$\mathcal{X}$D achieves performance comparable to or better than previous state-of-the-art and baseline methods in single-view 3D object generation, few-view 3D scene reconstruction, single-view 4D generation, and single/multi-view 4D generation.

## 2 RELATED WORK

**3D Generation.** Before the emergence of large-scale 3D data, early works (Melas-Kyriazi et al., 2023; Poole et al., 2022) distill the knowledge from 2D diffusion models for text- and image-based 3D generation. Later, with the development of 3D data, 3D generation has been mainly explored in two directions: multi-view priors and feed-forward models. Multi-view priors (Shi et al., 2023; Liu et al., 2023b;c; Long et al., 2024; Gao et al., 2024) generate multi-view images and other features (*e.g.*, normal maps and depths) based on camera embeddings, and then train a 3D representation using the generated samples or distill from generative priors. Feed-forward models (Hong et al., 2023; Liu et al., 2024b; 2023a; Tang et al., 2024; Hu et al., 2024) directly predict NeRF (Hong et al., 2023), 3D Gaussians (Tang et al., 2024; Tochilkin et al., 2024), or meshes (Liu et al., 2023a; 2024b) from single or multi-view images. Compared to multi-view priors, feed-forward models are more efficient but produce lower quality results. In this paper, we follow the paradigm of multi-view priors.

**4D Generation.** Similar to 3D generation, early 4D generation works (Zhao et al., 2023; Singer et al., 2023; Ling et al., 2024; Ren et al., 2023) distill 2D video generation models into 4D representation. Due to the variability and complexity of multi-view videos, these methods typically require long optimization time. Later, by leveraging animated 3D mesh data (Deitke et al., 2024), researchers render multi-view videos and use them to train 4D diffusion models (Liang et al., 2024) and feed-forward models (Ren et al., 2024). Although these models achieve good multi-view and video quality, they only focus on object-centric synthetic scenarios rather than entire scenes. This limitation is due to the lack of scene-level 4D data and the requirement for both multi-view static and dynamic information in these models. In this paper, we introduce a large-scale dataset of scene-level 4D data and address the challenge of general 4D generation.

**Camera-controlled Video Generation.** In the real world, videos contain not only object motion but also camera movement. Therefore, controlling camera movement in video generation has also garnered attention in the community (Wang et al., 2024b; Xu et al., 2024; He et al., 2024). MotionCtrl (Wang et al., 2024b) and CameraCtrl (He et al., 2024) introduce a branch to encode camera information from multi-view 3D data and integrate it into a frozen video generation model. However, due to the limitations of this integration approach, these methods cannot generate videos that align well with the camera pose. CamCo (Xu et al., 2024) annotates some 4D data similar to ours and fine-tunes the entire video generation model using this data. However, due to limitations in camera pose quality and diversity, CamCo struggles to handle large camera movements.

## 3 CAMVID-30K

The lack of large-scale 4D scene data limits the development of dynamic 3D tasks, including but not limited to 4D generation, dynamic camera pose estimation, and controllable video generation. To address this, we introduce a high-quality 4D dataset in this paper. First, we estimate the camera poses

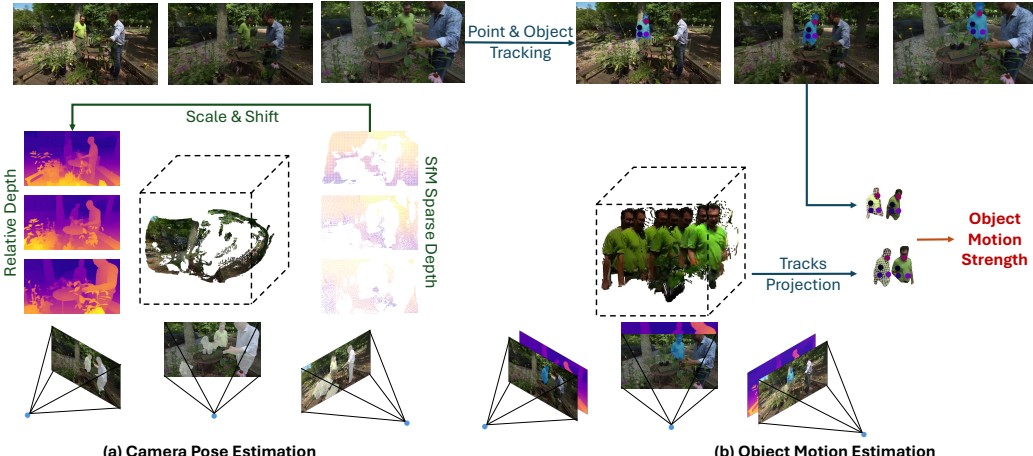

Figure 2: **The pipeline for CamVid-30K data curation**, including (a) camera pose estimation and (b) object motion estimation. We first leverage mask-based SfM (masks are overlaid to images in (a) for visualization) to estimate camera pose and reconstruct 3D point clouds of static parts. Then relative depth is aligned with the sparse depth and project the tracking keypoints to consecutive frame for object motion estimation.

using a Structure-from-Motion (SfM) based method, then filter out data without object movement using the proposed motion strength. The pipeline is illustrated in Fig. 2.

## 3.1 CAMERA POSE ESTIMATION

The camera pose estimation is based on SfM, which reconstructs 3D structure from their projections in a series of images. SfM involves three main steps: (1) feature detection and extraction, (2) feature matching and geometric verification, and (3) 3D reconstruction and camera pose estimation. In the second step, the matched features must be on the **static** part of the scene. Otherwise, object movement will be interpreted as camera movement during feature matching, which can impair the accuracy of camera pose estimations.

To address this, Particle-SfM (Zhao et al., 2022) separates moving objects from the static background using a motion segmentation module, and then performs SfM on the static part to estimate camera poses. However, it is extremely difficult to accurately detect moving pixels when the camera itself is moving, and we empirically observe that the motion segmentation module in Zhao et al. (2022) lacks sufficient generalization, leading to false negatives and incorrect camera poses. To obtain accurate camera poses, it is essential to segment **all** moving pixels. In this case, a *false positive* error is more acceptable than a false negative. To achieve this, we use an instance segmentation model (Cheng et al., 2022) to greedily segment all pixels that might be moving. The instance segmentation model is far more generalizable than the motion segmentation module in Zhao et al. (2022), particularly on training categories. After segmenting the potentially moving pixels, we estimate the camera pose with Particle-SfM (Zhao et al., 2022) to obtain camera information and sparse point clouds (Fig. 2(a)).

## 3.2 OBJECT MOTION ESTIMATION

**Unravel Camera and Object Motion.** While instance segmentation can accurately separate objects from backgrounds, it cannot determine whether the object itself is moving, and static objects negatively impact motion learning. Thus, we introduce motion strength to identify true object motion and filter out videos with only static objects.

Since camera movement and object motion are both present in videos, 2D-based motion estimation methods (*e.g.*, optical flow) cannot accurately represent true object motion. There are two ways to capture true object motion: by measuring motion in 3D space or by projecting motion in videos to the same camera. Both approaches require depth maps aligned with the camera pose scale. The

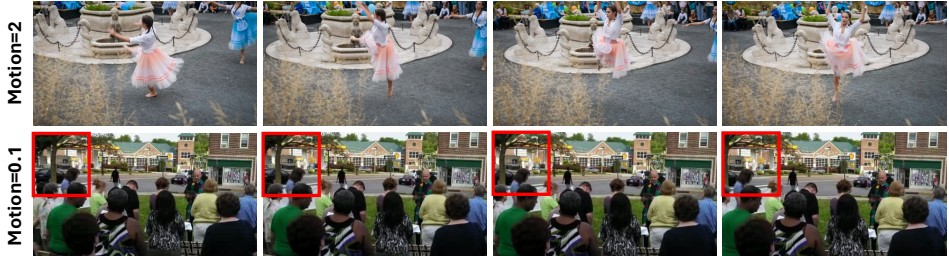

Figure 3: **Examples for object motion estimation.** The motion strength is multiplied by 100. In the first example, the girl is dancing, together with the camera moving. In the second example, the camera is zooming in (red rectangle for better illustration) but the object is static. In this case, the motion strength is much smaller.

sparse depth map can be obtained by projecting 3D point clouds $P_{\text{world}}$ onto the camera view:

$$P_{\text{camera}} = R \cdot P_{\text{world}} + t, \qquad (u, v, 1)^T = K \cdot (X_c/Z_c, Y_c/Z_c, 1)^T, \tag{1}$$

where $P_{\text{camera}} = (X_c, Y_c, Z_c)$ denotes the coordinates of point cloud $c$ in the camera space. $R$ and $t$ denote the rotation and translation to transform from world space to camera space. $K$ is the camera intrinsics. With the projection equation, the depth value $d_{\text{SfM}}$ on the image pixel $(u, v)$ can be obtained by $d_{\text{SfM}}(u, v) = Z_c$.

As shown in Fig. 2(a), since only features in the static parts are matched during 3D reconstruction, we can only obtain sparse point clouds for the static regions. However, depth information in the dynamic parts is crucial for estimating motion. To address this, we leverage a pre-trained relative monocular depth estimation model (Yang et al., 2024) to predict the relative depth of each frame $d_{\text{rel}} \in [0, 1]$. We then apply a scale factor $\alpha$ and a shift $\beta$ to align it with the SfM sparse depth:

$$\alpha = \text{median}(d_{\text{SfM}})/\text{median}(d_{\text{rel}}), \qquad \beta = \text{median}(d_{\text{SfM}} - \alpha \cdot d_{\text{rel}}),$$
$$d_{\text{aligned}} = \alpha * d_{\text{rel}} + \beta, \tag{2}$$

where $\text{median}(\cdot)$ denotes the median value, and $d_{\text{aligned}}$ is the dense depth map aligned with SfM depth scale.

**Object Motion Field.** With the aligned depth $d_{\text{align}}$, we can project dynamic objects in a frame into 3D space, providing a straightforward way to measure object motion. As shown in Fig. 2(b), if the object (*e.g.*, the man in the green shirt) is moving, there will be displacement in the projected 3D point clouds. However, since SfM operates up to a scale, measuring motion directly in 3D space can lead to magnitude issues. Therefore, we project the dynamic objects into adjacent views and estimate the object motion field.

Specifically, we first need to find matching points in the 2D video. Instead of using dense representations like optical flow, we sample keypoints for each object instance and use video object segmentation (Cheng et al., 2023) and keypoint tracking (Doersch et al., 2023) in 2D videos to establish matching relationships. Each keypoint is then projected into adjacent frames. The keypoint $(u_i, v_i)^T$ in the $i$-th frame is first back-projected into world space to obtain the 3D keypoint $kp_i$:

$$kp_i = R_i^{-1} \left( Z_i \cdot K^{-1} \cdot (u_i, v_i, 1)^T - t_i \right), \tag{3}$$

where $Z_i = d_{\text{aligned}}(u_i, v_i)$ is the depth value in the aligned dense depth map. Then the 3D keypoint is projected to $j$-th frame with the projection equation (Eq. 1) to obtain the 2D projected keypoint $(u_{ij}, v_{ij})^T$. Similar to optical flow, we represent the displacement of each 2D keypoint on the second camera view as object motion field:

$$(\Delta u_{ij}, \Delta v_{ij})^T = ((u_j - u_{ij})/W, (v_j - v_{ij})/H)^T, \tag{4}$$

where $H$ and $W$ denotes image height and width.

With the motion field for each object, we can estimate the global movement of an object by averaging the absolute magnitude of the motion field. For each video, the motion strength $\gamma$ is represented by the maximum movement value among all the objects. As shown in Fig. 3, when the camera is moving while the object remains static (second example), the motion strength is significantly smaller compared to videos with object motion. Using motion strength, we further filter out data that lacks obvious object movement. The motion strength value also serves as a good indicator of the scale of object movement, which is used in the temporal layer to enable better motion control (Sec. 4.1).

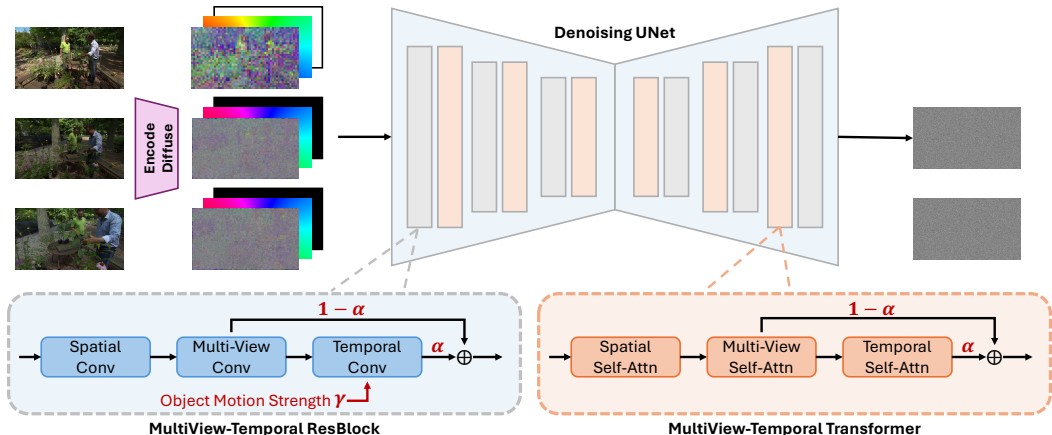

Figure 4: **The framework of Gen$\mathcal{X}$D.** We leverage mask latent conditioned diffusion model to generate 3D and 4D samples with both camera (colorful map) and image (binary map) conditions. In addition, multiview-temporal modules together with $\alpha$-fusing are proposed to effectively disentangle and fuse multiview and temporal information.

## 4 GEN$\mathcal{X}$D

### 4.1 GENERATIVE MODEL

Since most scene-level 3D and 4D data are captured via videos, these data lack explicit representations (*e.g.*, meshes). Therefore, we adopt an approach that generates images aligned with spatial camera poses and temporal timesteps. Specifically, we incorporate the latent diffusion model (Rombach et al., 2022) into our framework, introducing additional multiview-temporal layers, including multiview-temporal ResBlocks and multiview-temporal transformers, to disentangle and fuse 3D and temporal information.

**Mask Latent Conditioned Diffusion Model.** Latent diffusion model (LDM) (Rombach et al., 2022) is leveraged in Gen$\mathcal{X}$D to generate images of different camera viewpoint and time together. LDM first encode an image/video into a latent code $z$ with VAE (Kingma, 2013) and diffuse the latent with gaussian noise $\epsilon$ to obtain $z_t$. Then a denoising model $\epsilon_\theta(\cdot)$ is leveraged to estimate the noise and reverse the diffusion process with conditions:

$$L_{\text{LDM}} := \mathbb{E}_{\mathcal{E}(x), \epsilon \sim \mathcal{N}(0,1), t}\left[\|\epsilon - \epsilon_\theta(z_t, t, c)\|_2^2\right], \tag{5}$$

where $c$ is the condition used for controllable generation, which is commonly text or image.

Gen$\mathcal{X}$D generates multi-view images and videos with camera pose and reference image, and thus it requires both camera and image conditions. Camera conditions are independent for each image, either conditioned or targeted. Therefore, it is easy to append it to each latent. Here, we opt for Plücker ray (Plücker, 1828) as camera condition:

$$\mathbf{r} = \langle \mathbf{d}, \mathbf{o} \times \mathbf{d} \rangle \in \mathbb{R}^6 \tag{6}$$

where $\mathbf{o} \in \mathbb{R}^3$ and $\mathbf{d} \in \mathbb{R}^3$ denote the camera center and the ray direction from camera center to each image pixel, respectively. Therefore, Plücker ray is a dense embedding encoding not only the pixel information, but also the camera pose and intrinsic information, which is better than global camera representation.

The reference image condition is more complex. Gen$\mathcal{X}$D aims to conduct 3D and 4D generation with both single and multiple input views. The single view generation has less requirement while the multi-view generation has more consistent results. Therefore, combining single and multi-view generation will lead to better real-world application. However, previous works (Blattmann et al., 2023; Voleti et al., 2024; Liu et al., 2023b) condition images by concatenating condition latent to the target latents and by incorporating CLIP image embedding (Radford et al., 2021) via cross attention. The concatenation way requires to change the channel of the model, which is unable to process arbitrary input views. The CLIP embedding can support multiple conditions. However, both ways

cannot model the positional information of multiple conditions and model the information among the input views. In view of the limitations, we leverage the mask latent conditioning (Gao et al., 2024; Jain et al., 2024) for image conditions. As shown in Fig. 4, after encoding with VAE encoder, the forward diffusion process is applied to the target frames (2nd and 3rd frame), leaving the condition latent (1st frame) as usual. Then the noise on the two frames are estimated by denoising model and removed by the backward process.

The mask latent conditioning has three main benefits. First, model can support any input views without modification on the parameters. Second, for sequence generation (multi-view images or video), we do not need to constraint the position of the condition frame since the condition frame keeps its position in the sequence. In contrast, many works (Blattmann et al., 2023; Xu et al., 2024; Chen et al., 2023) requires the condition image at a fixed position in the sequence (commonly the first frame). Third, without the conditioning embedding from additional models (Radford et al., 2021), the cross attention layers used to integrate conditioning embedding can be removed, which will greatly reduce the number of model parameters. To this end, we leverage mask latent conditioning approach for Gen$\mathcal{X}$D.

**MultiView-Temporal Modules.** As Gen$\mathcal{X}$D aims to generate both 3D and 4D samples within a single model, we need to disentangle the multi-view information from the temporal information. We model these two types of information in separate layers: the multi-view layer and the temporal layer. For 3D generation, no temporal information is considered, while both multi-view and temporal information are required for 4D generation. Therefore, as illustrated in Fig. 4, we propose an $\alpha$-**fusing** strategy for 4D generation. Specifically, we introduce a learnable fusing weight, $\alpha$, for 4D generation, with $\alpha$ set to 0 for 3D generation. Using the $\alpha$-fusing strategy, Gen$\mathcal{X}$D can preserve the multi-view information in the multi-view layer for 3D data while learning the temporal information from 4D data.

$\alpha$-fusing can effectively disentangle the multi-view and temporal information. However, the motion is less controllable without any cues. Video generation models (Zhou et al., 2022; Blattmann et al., 2023) use FPS or motion id to control the magnitude of motion without considering the camera movement. Thanks to the motion strength $\gamma$ in CamVid-30K, we can effectively represent the object motion. Since the motion strength is a constant, we combine it with the diffusion timestep and add it to the temporal resblock layer, as illustrated in MultiView-Temporal ResBlock of Fig. 4. With the multiview-temporal modules, Gen$\mathcal{X}$D can effectively conduct both 3D and 4D generation.

### 4.2 GENERATION WITH 3D REPRESENTATION

Gen$\mathcal{X}$D can generate images with different viewpoints and timesteps using one or several condition images. However, to render arbitrary 3D-consistent views, we need to lift the generated samples into a 3D representation. Previous works (Wu et al., 2024b; Shi et al., 2023; Zhao et al., 2023) commonly optimize 3D representations by distilling knowledge from generative models. Since Gen$\mathcal{X}$D can generate high-quality and consistent results, we directly use the generated images to optimize the 3D representation. Specifically, we utilize 3D Gaussian Splatting (3D-GS) (Kerbl et al., 2023) and Zip-NeRF (Barron et al., 2023) for 3D generation, and 4D Gaussian Splatting (4D-GS) (Wu et al., 2024a) for 4D generation. More details can be found in Appendix. H.

## 5 EXPERIMENT

### 5.1 EXPERIMENTAL SETUP

**Datasets.** Gen$\mathcal{X}$D is trained with the combination of 3D and 4D datasets. For 3D datasets, we leverage five datasets with camera pose annotation: Objaverse (Deitke et al., 2023), MVImageNet (Yu et al., 2023), Co3D (Reizenstein et al., 2021), Re10K (Zhou et al., 2018) and ACID (Liu et al., 2021). Objaverse is a synthetic dataset with meshes, and we render the 80K subset (Tang et al., 2024) from 12 views following (Liu et al., 2023b). MVImageNet and Co3D are video data recording objects in 239 and 50 categories, respectively. Re10K and Acid are video data that record real-world indoor and outdoor scenarios. For 4D datasets, we leverage the synthetic data Objaverse-XL-Animation (Deitke et al., 2024; Liang et al., 2024) and our CamVid-30K. For the Objaverse-XL-Animation, we use the subset filtered by Liang et al. (2024), and re-render the depth and images by adding noise to the oribit camera trajectory. With the ground truth depth, we estimate the object

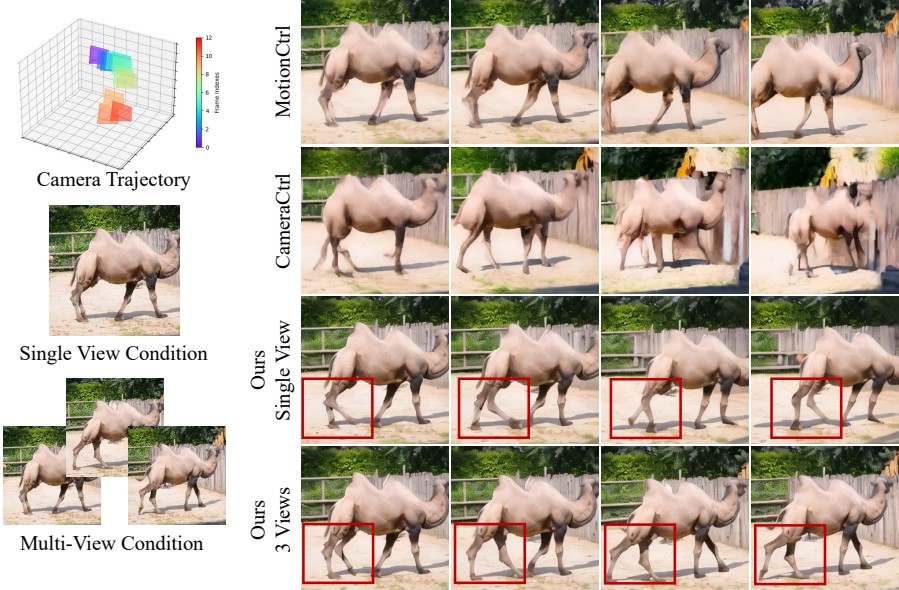

Figure 5: **Qualitative comparison with camera conditioned video generation methods.** Gen$\mathcal{X}$D can generate video well-aligned with camera trajectory and containing realistic object motion. (Please refer to supplementary video for better illustration.)

Table 2: **4D scene generation.**

| Method | FID ↓ | FVD ↓ |
|---|---|---|
| MotionCtrl (Wang et al., 2024b) | 118.14 | 1464.08 |
| CameraCtrl (He et al., 2024) | 138.64 | 1470.59 |
| Gen$\mathcal{X}$D (Single View) | 101.78 | 1208.93 |
| Gen$\mathcal{X}$D (3 Views) | **55.64** | **490.50** |

Table 3: **4D object generation.**

| Method | Time ↓ | CLIP-I ↑ |
|---|---|---|
| Zero-1-to-3-V (Liu et al., 2023b) | 4 hrs | 79.25 |
| RealFusion-V (Melas-Kyriazi et al., 2023) | 5 hrs | 80.26 |
| Animate124 (Zhao et al., 2023) | 7 hrs | 85.44 |
| Gen$\mathcal{X}$D (Single View) | **4 min** | **90.32** |

motion strength as in Sec. 3.2, and then filter out data without obvious object motion. Finally, we get 44K synthetic data from Objaverse-XL-Animation and 30K real-world data from CamVid-30K.

**Implementation Details.** Gen$\mathcal{X}$D is partially initialized from Stable Video Diffusion (SVD) pre-trained model (Blattmann et al., 2023) for fast convergence. Specifically, both the multi-view layer (multi-view convolution and multi-view self-attention) and temporal layer (temporal convolution and temporal self-attention) are initialized from the temporal layer in SVD, and the cross-attention layers in SVD are removed. Gen$\mathcal{X}$D is trained in three stages. We first train the UNet only with 3D data for 500K iteration and then fine-tune it with both 3D and 4D data for 500K iterations in single view mode. Finally, Gen$\mathcal{X}$D is trained with both single view and multi-view mode with all the data for 500K iteration. The model is trained on 32 A100 GPUs with batch size 128 and resolution 256×256. AdamW (Loshchilov & Hutter, 2019) optimizer with learning rate $5 \times 10^{-5}$ is adopted. In the first stage, data are center cropped to square. In the final stage, we make the images square by either center crop or padding, leading to Gen$\mathcal{X}$D working well on different image ratio.

## 5.2 4D GENERATION

**4D Scene Generation.** In this setting, videos with both object and camera movement are required for evaluation. Therefore, we introduce the Cam-DAVIS benchmark for 4D evaluation. Specifically, we use our annotation pipeline proposed in Sec. 3 to get the camera poses for videos in DAVIS (Perazzi et al., 2016) dataset. Then we filter the data and get 20 videos with both accurate camera poses and obvious object movement. The camera trajectories of Cam-DAVIS are out-of-distribution from the training data and thus are good evaluation for the robustness to the camera movement.

We compare Gen$\mathcal{X}$D with the open-sourced camera conditioned video generation methods, MotionCtrl (Wang et al., 2024b) and CameraCtrl (He et al., 2024), on FID (Heusel et al., 2017) and FVD (Unterthiner et al., 2018) evaluation metrics. We use Stable Video Diffusion (Blattmann et al.,

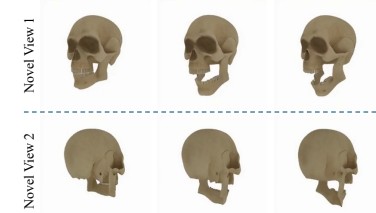

| 3D-GS | Gen𝒳D (3D-GS) | Zip-NeRF | Gen𝒳D (Zip-NeRF) |

Figure 6: **Qualitative comparison of few-view 3D reconstruction.**

Figure 7: **Multi-view videos generated by Gen𝒳D.**

Table 4: **Quantitative comparison of few-view 3D reconstruction** on both in-distribution (Re10K) and out-of-distribution (LLFF) datasets.

| Method | Re10K | | | LLFF | | |
|---|---|---|---|---|---|---|
| | PSNR↑ | SSIM↑ | LPIPS↓ | PSNR↑ | SSIM↑ | LPIPS↓ |
| Zip-NeRF (Barron et al., 2023) | 20.58 | 0.729 | 0.382 | 14.26 | 0.327 | 0.613 |
| **Zip-NeRF + Gen𝒳D** | **25.40** | **0.858** | **0.223** | **19.39** | **0.556** | **0.423** |
| 3D-GS (Kerbl et al., 2023) | 18.84 | 0.714 | 0.286 | 17.35 | 0.489 | 0.335 |
| **3D-GS + Gen𝒳D** | **23.13** | **0.808** | **0.202** | **19.43** | **0.554** | **0.312** |

2023) as the base model for both previous methods and generate the videos with the camera trajectory and the first frame conditions. As shown in Tab. 2, using first view as condition, Gen𝒳D outperforms CameraCtrl and MotionCtrl in terms of both metrics significantly. In addition, with 3 views as conditions (first, middle and last frames), Gen𝒳D outperforms previous works by a large margin. Such results demonstrate the strong generalization ability of Gen𝒳D on 4D generation. In Fig. 5, we compare the qualitative results of the three methods. In this example, MotionCtrl cannot generate obvious object motion and the video generated by CameraCtrl is neither 3D nor temporal consistent. Instead, our single view conditioned model can generate smooth and consistent 4D video. With 3 conditioning views, Gen𝒳D can generate quite realistic results.

**4D Object Generation.** We evaluate the performance on 4D object generation following Zhao et al. (2023). Since Gen𝒳D only leverages image condition instead of image-text condition as Animate124 (Zhao et al., 2023), we compare the optimization time and CLIP image similarity in Tab. 3. Instead of optimizing dynamic NeRF with score distillation sampling (SDS) (Poole et al., 2022), Gen𝒳D directly generates 4D videos of the orbit camera trajectory and uses such videos to optimize the 4D-GS (Wu et al., 2024a). With the effective generated videos and 4D representation, our method can be much faster than Animate124 (4 minutes *v.s.* 7 hours). In addition, the semantic drift problem mentioned in Zhao et al. (2023) is well addressed in Gen𝒳D since we use the condition image for 4D generation. The results on 4D scene and object generation demonstrate the superiority of Gen𝒳D in generating 3D and temporal consistent 4D videos. Furthermore, Gen𝒳D is able to generate synchronized multi-view videos from a fixed view video. One qualitative example is shown in Fig. 7, and the implementation details are provided in Appendix. D.

### 5.3 3D GENERATION

**Few View 3D Generation.** For few-view 3D reconstruction setting, we evaluate Gen𝒳D on both in-distribution (Re10K (Zhou et al., 2018)) and out-of-distribution (LLFF (Mildenhall et al., 2019)) datasets. We select 10 scenes from Re10K and all the 8 scenes in LLFF, and 3 views in each scene are used for training. The performance is evaluated with PSNR, SSIM and LPIPS metrics on the rendered test views. As a generative model, Gen𝒳D can generate additional views from sparse input views and improve the performance of any reconstruction method. In this experiment, we leverage two baseline methods: Zip-NeRF (Barron et al., 2023) and 3D-GS (Kerbl et al., 2023). The two baselines are methods for many-view reconstruction, and thus we adjust the hyperparameter for better few-view reconstruction (more details in Appendix. H). As shown in Tab. 4, both Zip-NeRF and 3D-GS can be improved with the generated images from Gen𝒳D, and the improvement is more significant with the Zip-NeRF baseline. Specifically, the PSNR on Re10K (in-distribution) and LLFF (out-of-distribution) are increased by 4.82 and 5.13. The qualitative comparison is illustrated

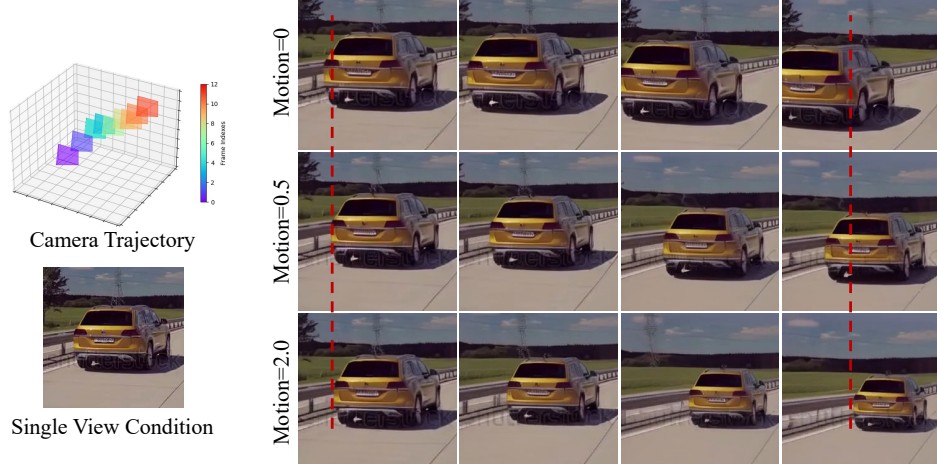

Figure 8: **Qualitative evaluation on the influence of motion strength.** (Please refer to supplementary video for better illustration.)

Table 5: **Ablation studies on motion disentangle.**

| Method | Re10K | | | LLFF | | | Cam-DAVIS | |
|---|---|---|---|---|---|---|---|---|
| | PSNR↑ | SSIM↑ | LPIPS↓ | PSNR↑ | SSIM↑ | LPIPS↓ | FID↓ | FVD↓ |
| w.o. Motion Disentangle | 20.75 | 0.635 | 0.362 | 16.89 | 0.397 | 0.560 | 122.73 | 1488.47 |
| Gen$\mathcal{X}$D | **22.96** | **0.774** | **0.341** | **17.94** | **0.463** | **0.546** | **101.78** | **1208.93** |

in Fig. 6. With the generated views, the floaters and blurs are reduced in the reconstructed scene. We also evaluate the performance on single view generation setting in Appendix. B.

## 5.4 ABLATION STUDY

In this section, we conduct the ablation study of multiview-temporal modules. The ablation study is evaluated on the quality of the generated diffusion samples on few view 3D and single view 4D generation settings (Tab. 5). More ablation studies are conducted in Appendix. G

**Motion Disentangle ($\alpha$-fusing).** The camera movement and object motion are entangled in 4D data. To enable high quality generation in both 3D and 4D, Gen$\mathcal{X}$D introduces multiview-temporal modules (Sec. 4.1) to learn the multi-view and temporal information separately, and then fuse them together with $\alpha$-fusing. For 3D generation, the $\alpha$ is set to 0 to bypass the temporal module while the $\alpha$ is learned during training for 4D generation. Removing the $\alpha$-fusing will result in all 3D and 4D data passing through temporal modules, which will result in the model being unable to disentangle object motion from camera movement. The failure of disentanglement will adversely affect both 3D and 4D generation (Tab. 5).

**Effectiveness of Motion Strength $\gamma$.** The motion strength can be used to effectively control the magnitude of the object's movement. As shown in the second to last row of Fig. 8, increasing the motion strength can increase the speed of the car. As a result of these observations, we can conclude that it is important to learn object motion and that the object motion field and motion strength in our data curation pipeline can accurately represent true object motion.

## 6 CONCLUSION

In this paper, we investigate the general 3D and 4D generation with diffusion models. To enhance the learning of 4D generation, we first propose a data curation pipeline to annotate camera and object movement in the videos. Equipped with the pipeline, the largest real-world 4D scene dataset, CamVid-30K, is introduced in this paper. Furthermore, leveraging the large-scale datasets, we propose Gen$\mathcal{X}$D to handle general 3D and 4D generation. Gen$\mathcal{X}$D utilize multiview-temporal modules to disentangle camera and object movement and is able to support any number of input condition views by mask latent conditioning. Gen$\mathcal{X}$D can handle versatile applications and can achieve comparable or better performance in all settings with one model.

ACKNOWLEDGEMENTS

This work was partially carried out during Yuyang Zhao's internship at Microsoft. This work is also supported by the National Research Foundation (NRF) Singapore, under its NRF-Investigatorship Programme (Award ID. NRF-NRFI09-0008).

Additionally, Yuyang would also like to express his deepest gratitude to Dr. Peng Qi for her unwavering love, patience, and inspiration throughout this journey and would like to ask her: Peng, will you marry me?

ETHICS STATEMENT

In this paper, we introduce a 4D dataset, CamVid-30K, and a generative model for general 3D and 4D generation. CamVid-30K is curated from existing public video datasets (Nan et al., 2024; Bain et al., 2021; Miao et al., 2022), and we additionally estimate the camera poses and object motion. CamVid-30K adheres to the licenses and agreements of the original video datasets, and it does not raise any new ethical concerns. While we ensure compliance with the licenses of the curated datasets and advocate for responsible use, the ability to generate realistic images and videos from various viewpoints raises risks related to misinformation and privacy violations. We encourage the development of tools for detecting misuse, along with responsible dissemination of the dataset and model, to balance innovation with ethical considerations.

REPRODUCIBILITY STATEMENT

The experiments in our paper mainly include the training of Gen$\mathcal{X}$D and generation with 3D representation in different settings. In Sec. 5.1, we describe the 3D and 4D datasets used to train the diffusion model, together with the training configurations. In Appendix. H, we introduce the backbone models and the implementation details for generation with 3D representation in each setting. Our curated 4D dataset, CamVid-30K, and Gen$\mathcal{X}$D model will be made publicly available.

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

APPENDIX

## A MORE QUALITATIVE RESULTS

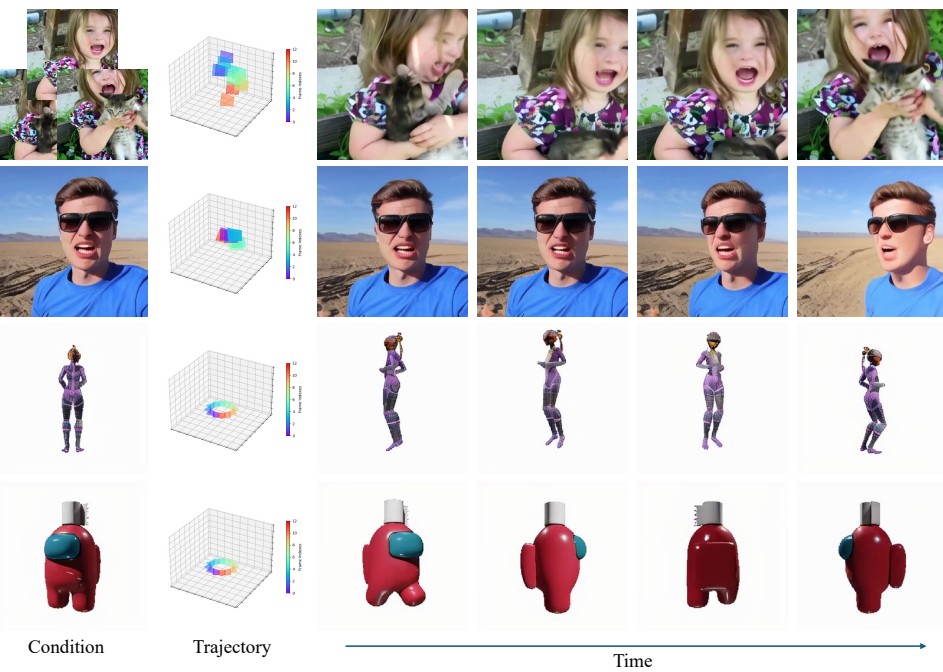

Figure 9: The visualization of the generated 4D videos. (Please refer to supplementary video for better illustration.)

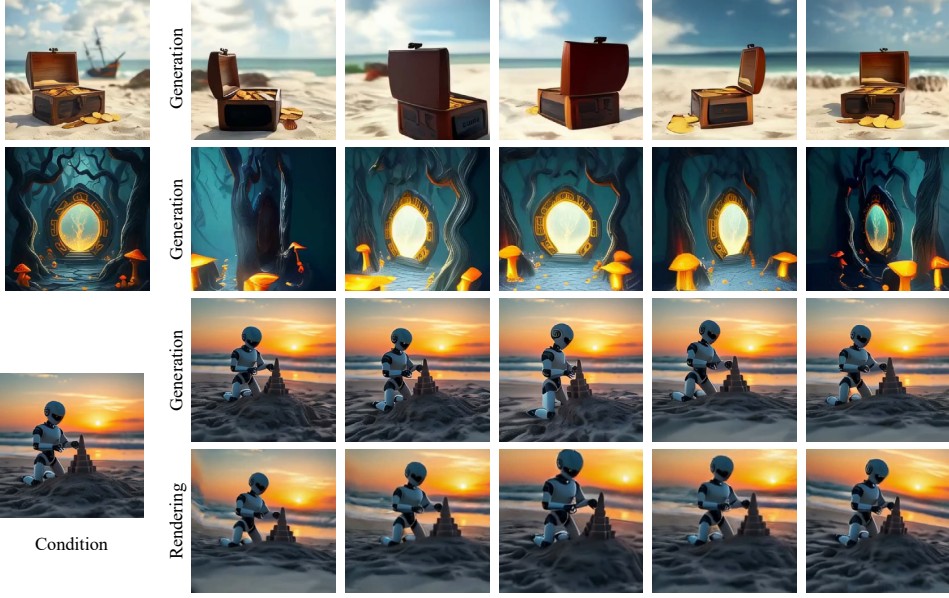

Figure 10: The visualization of the 3D generation results. The condition images are generated by FLUX.1. The first three rows are the generated samples from Gen𝒳D following camera trajectory, and the last row is the renderings from 3D-GS model trained with the generated samples. (Please refer to supplementary video for better illustration.)

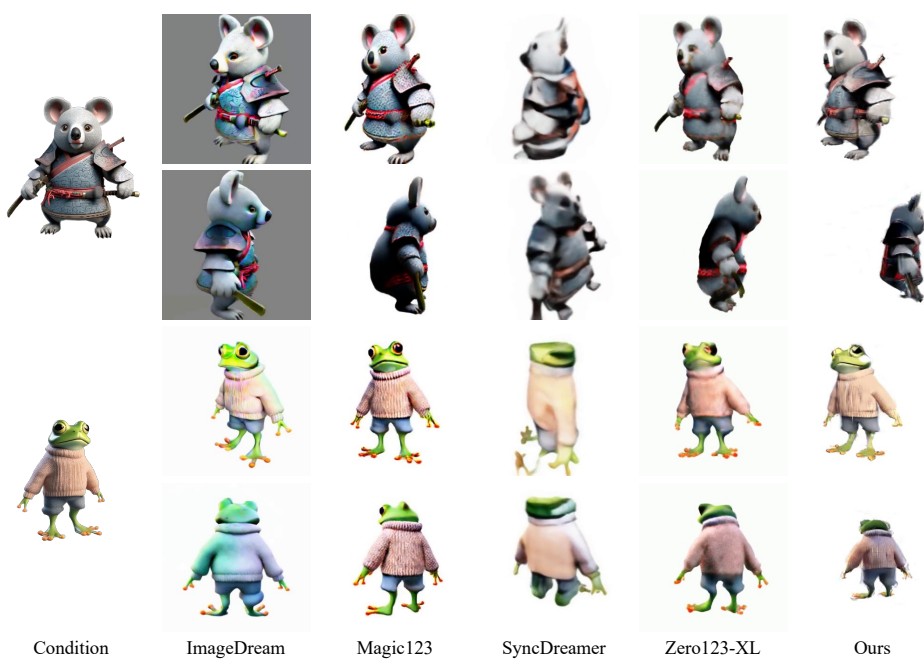

| Condition | ImageDream | Magic123 | SyncDreamer | Zero123-XL | Ours |

Figure 11: Qualitative comparison on single view 3D generation.

Table 6: Quantitative comparison of image-to-3D generation on examples from Wang & Shi (2023).

| Method | Model Type | Time (min) ↓ | CLIP-I ↑ |
|---|---|---|---|
| ImageDream (Wang & Shi, 2023) | 3D | 120 | 83.77 |
| One2345++ (Liu et al., 2024a) | 3D | **0.75** | 83.78 |
| IM-3D (Melas-Kyriazi et al., 2024) | 3D | 3 | **91.40** |
| Gen$\mathcal{X}$D (ours) | 3D&4D | 2 | 84.75 |

In this section, we show more qualitative results in general 3D and 4D generation with Gen$\mathcal{X}$D. In Fig. 9, Gen$\mathcal{X}$D can generate 3D and temporal consistent videos following the camera trajectory in with one or multiple condition images. In addition, Gen$\mathcal{X}$D can also achieve high quality 3D generation with in-the-wild images. In Fig. 10, we use FLUX.1[1] to generate condition images and use Gen$\mathcal{X}$D to generate multi-view images following orbit and forward-facing trajectories. Furthermore, the generated samples are 3D consistent and we can lift them to 3D representations (*e.g.*, 3D-GS). The last row of Fig. 10 is the renderings from 3D-GS model, which is optimized from the generated samples in the third row.

## B SINGLE VIEW 3D GENERATION

We evaluate Gen$\mathcal{X}$D on the object-centric single view 3D generation on ImageDream benchmark (Wang & Shi, 2023) and GSO dataset (Downs et al., 2022). First, we use the examples in Wang & Shi (2023) and calculate the evaluation metric with 7 views in the orbit trajectory. The similarity between CLIP image embedding of the reference image and rendered views are adopted as evaluation metric. As shown in Tab. 6, although our performance is slightly worse than IM-3D (Melas-Kyriazi et al., 2024), Gen$\mathcal{X}$D achieves better performance than other baselines (Liu et al., 2024a; Wang & Shi, 2023) with a fast generation speed. In addition, Gen$\mathcal{X}$D can handle scene-level generation and 4D generation, which cannot be achieved by other methods. Furthermore, we only use basic 3D-GS model and train with the generated views. Approaches proposed by previous works (Tang et al., 2023; Melas-Kyriazi et al., 2024) to improve the 3D quality can still be used. In Fig. 11, we compare the rendering results from Gen$\mathcal{X}$D with several previous meth-

---

[1]https://huggingface.co/black-forest-labs/FLUX.1-schnell

ods (Qian et al., 2023; Liu et al., 2023b; Wang & Shi, 2023; Liu et al., 2023c). The results of other methods are obtained from Wang & Shi (2023). Compared with previous methods, Gen$\mathcal{X}$D use the generated 3D consistent samples to optimize the 3D-GS model, instead of using distillation techniques (Poole et al., 2022). Therefore, Gen$\mathcal{X}$D can well handle the over-saturated and Janus problem faced by other methods.

Second, we further evaluate the 3D generation ability of Gen$\mathcal{X}$D on Google Scanned Objects (GSO) dataset. Here, we leverage the benchmark in Kong et al. (2024), which evaluates the novel view synthesis (NVS) and 3D reconstruction accuracy on 30 samples in GSO. For NVS evaluation, 25 views are generated from the generative model and are used to calculate the PSNR, SSIM, and LPIPS with the ground truth view. For 3D reconstruction evaluation, 36 frames are generated and used to optimize NeuS. Chamfer distance and volume IoU of the exported mesh are used as the metrics. As shown in Table. 7, Gen$\mathcal{X}$D achieves the best performance on both NVS and 3D reconstruction evaluation. In addition, the compared method cannot work well on scene-level 3D generation, while Gen$\mathcal{X}$D can handle both object-level and scene-level generation on 3D and 4D.

Table 7: Evaluation of 3D object generation on GSO-30.

| Method | NVS | | | 3D Reconstruction | |
|---|---|---|---|---|---|
| | PSNR↑ | SSIM↑ | LPIPS↓ | Chamfer Dist. ↓ | Volume IoU ↑ |
| Zero123 | 18.51 | 0.856 | 0.127 | — | — |
| Zero123-XL | 18.93 | 0.856 | 0.124 | — | — |
| EscherNet | 20.24 | 0.884 | 0.095 | 0.0314 | 0.5974 |
| **Gen$\mathcal{X}$D (Ours)** | **21.08** | **0.888** | **0.094** | **0.0234** | **0.6770** |

## C    VIDEO-TO-4D EVALUATION

Following STAG4D (Zeng et al., 2024), we evaluate Gen$\mathcal{X}$D on video-to-4D benchmark proposed by Jiang et al. (2024). The performance is evaluated with the CLIP similarity, LPIPS, and FVD between the rendered video and the ground truth video. To realize video-to-4D, we first generate multi-view images for each video frame in our 3D generation mode. The multi-view images of the first frame are used to optimize 3D gaussian model first, and then we use the multi-view videos to optimize 4DGS (Wu et al., 2024a). After optimizing 4DGS, we conduct video refinement. Specifically, we render a video from 4DGS, add noise to the latent, and generate the video with Gen$\mathcal{X}$D in 4D generation mode. The generated video is used to further optimize 4DGS. The results are shown in Tab. 8. Gen$\mathcal{X}$D outperforms Consistent4D (Jiang et al., 2024), DreamGaussian4D (Ren et al., 2023) and STAG4D (Zeng et al., 2024) on all the evaluation metrics. In addition, Gen$\mathcal{X}$D achieves better CLIP similarity and LPIPS than SV4D (Xie et al., 2024), and comparable FVD with SV4D.

Table 8: Evaluation of video-to-4D object generation on Consistent4D. † denotes using the official open-source code to do the evaluation.

| Method | CLIP↑ | LPIPS↓ | FVD↓ |
|---|---|---|---|
| Consistent4D | 0.87 | 0.16 | 1133.44 |
| DreamGaussian4D[†] | 0.92 | 0.15 | 782.94 |
| STAG4D[†] | 0.92 | 0.13 | 963.61 |
| SV4D | 0.92 | 0.118 | **732.40** |
| **Gen$\mathcal{X}$D** | **0.93** | **0.113** | 733.25 |

## D    GENERATING MULTI-VIEW VIDEOS

Gen$\mathcal{X}$D has the capability to generate multi-view videos from a single video, as shown in Fig. 12. Specifically, we first generate multi-view images for each frame with our model in 3D generation mode to get the coarse multi-view videos. After that, for each video, we select $N$ frames ($N$=12

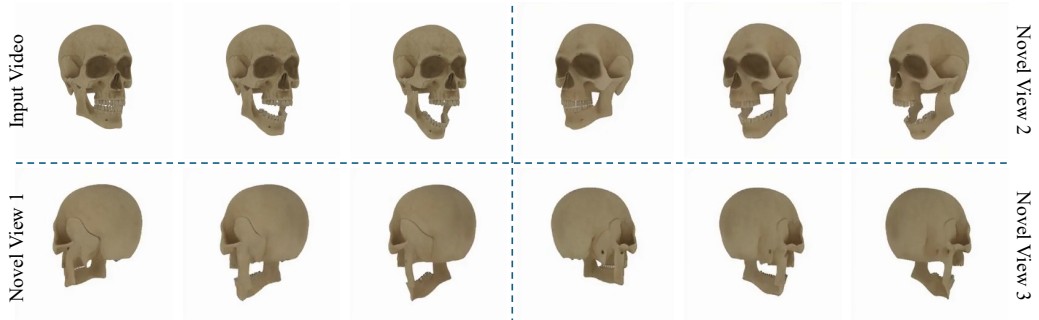

Figure 12: Qualitative results of multi-view videos.

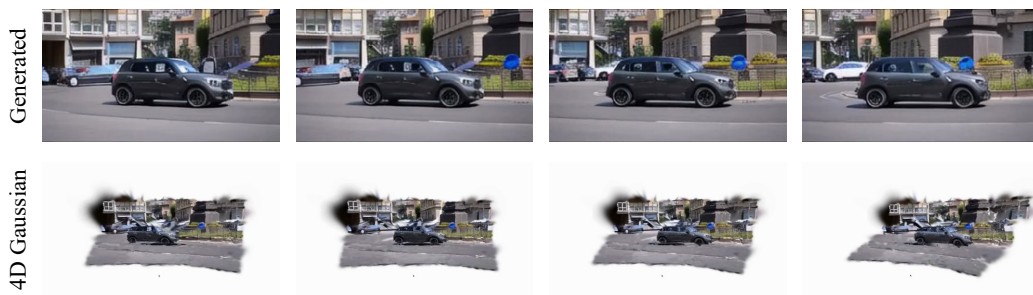

Figure 13: Qualitative results of 4D representation of scene-level 4D gneration.

aligned with our training setting) at once, take three of them as conditions, and add noise to the remaining frames. Then the video is fed into Gen$\mathcal{X}$D for denoising with 4D generation mode, obtaining the refined video. The refinement process is similar to video editing, which is widely adopted by 3D (Tang et al., 2023) and 4D (Ren et al., 2023) generation methods.

## E    SCENE-LEVEL 4D REPRESENTATION

Gen$\mathcal{X}$D can generate spatially and temporally consistent 4D videos, allowing them to be lifted to 4D representations. Results are shown in Fig. 13. Specifically, we used the dynamic gaussian representation from Wang et al. (2024a), which deforms 3D gaussians using basis trajectories. This representation is supervised by photometric, depth, and mask losses.

## F    CAMERA ACCURACY EVALUATION

In this section, we evaluate the camera accuracy of the generated video in terms of translation error, rotation error and the COLMAP success ratio. Specifically, COLMAP is applied to the generated results and the videos with all the frames registered are viewed as success. The normalized translation error and rotation error is calculated on the success samples. As shown in Table. 9, Gen$\mathcal{X}$D can achieves better performance on both in-distribution (Re10K) and out-of-distribution (DL3DV) data.

## G    ADDITIONAL ABLATION STUDIES

In this section, we further analyze the effectiveness of ray camera condition and the joint training of 3D and 4D generation.

**Camera Condition.** Gen$\mathcal{X}$D leverages Plücker ray as camera condition, which is a dense pixel-wise camera representation. Some previous works (Liu et al., 2023b; Sargent et al., 2024) convert the camera poses or spherical coordinate system to 1-dimension embeddings, and then integrate the

Table 9: Evaluation of camera controlled video generation on both in-distribution (Re10K) and out-of-distribution (DL3DV) dataset.

| Method | Re10K Trajactory | | | DL3DV Trajactory | | |
|---|---|---|---|---|---|---|
| | TransError.↓ | RotError.↓ | Success↑ | TransError.↓ | RotError.↓ | Success↑ |
| MotionCtrl | 0.0996 | 0.2884 | 0.54 | 0.1557 | 0.8054 | 0.57 |
| CameraCtrl | 0.0871 | 0.2730 | 0.62 | 0.1304 | 0.8251 | 0.75 |
| Gen$\mathcal{X}$D (Ours) | **0.0759** | **0.2341** | **0.73** | **0.1007** | **0.8019** | **0.89** |

Table 10: Ablation studies on camera conditioning scheme and joint training.

| Method | Re10K | | | LLFF | | | Cam-DAVIS | |
|---|---|---|---|---|---|---|---|---|
| | PSNR↑ | SSIM↑ | LPIPS↓ | PSNR↑ | SSIM↑ | LPIPS↓ | FID↓ | FVD↓ |
| Camera CA | 21.73 | 0.692 | 0.355 | 17.15 | 0.434 | 0.573 | 105.69 | 1331.62 |
| w.o. 3D Data | 16.38 | 0.604 | 0.465 | 14.98 | 0.400 | 0.587 | 107.74 | 1262.12 |
| w.o. 4D Data | 20.74 | 0.740 | 0.359 | 17.35 | 0.448 | 0.554 | 107.93 | 1240.57 |
| Gen$\mathcal{X}$D | **22.96** | **0.774** | **0.341** | **17.94** | **0.463** | **0.546** | **101.78** | **1208.93** |

condition into diffusion model with cross attention. As shown in the first row of Tab. 10, we convert the $3\times4$ camera extrinsics and the focal length to 1-dimension embeddings and use it with cross attention layer (Camera CA). Compared to Gen$\mathcal{X}$D, Camera CA performs worse on both 3D and 4D generation. In addition, due to the effectiveness of mask latent conditioned diffusion model, Gen$\mathcal{X}$D do not requires cross attention in the denoising U-Net, which is more efficient.

**Joint training of 3D and 4D generation.** Gen$\mathcal{X}$D combine both 3D and 4D data into modeling training to facilitate 3D and 4D generation. In Tab. G, we conduct ablation studies on the effectiveness of each type of data. 3D data contains more camera position variation and more accurate camera poses. Thus, removing 3D data impairs both 3D and 4D generation, especially the generated 3D samples which cannot well aligned with the camera poses. Instead, 4D data contains object motion together with spatial camera information. The lack of 4D data will impair the quality of the generated 3D samples. In addition, despite the performance drop on Cam-DAVIS is not significant in terms of FVD and FID, removing 4D data leads the model hardly generate object movement.

# H  DETAILS OF GENERATION WITH 3D REPRESENTATION

For few-view 3D reconstruction, we fit trajectories from training views and generate samples corresponding to the cameras on the trajectory. Then the training views and generated samples are used together to optimize the 3D representation (3D Gaussian Splatting (3D-GS) (Kerbl et al., 2023) and Zip-NeRF (Barron et al., 2023)). Since both of them are designed for many views reconstruction, we modify the hyperparameter to fit for few-shot setting. For Zip-NeRF, the width and depth of view-dependence network is set to 32 and 1 to avoid overfitting. The model is trained with batch size 262,144 ($64\times64\times64$) for 1k iterations. At each iteration, patch of $64\times64$ are rendered and supervised with photometric and LPIPS loss. For 3D-GS, the model is optimized for 10k iterations with photometric, SSIM and LPIPS loss. The initial point clouds are obtained from the input views for the baseline and all the generated views when combining it with Gen$\mathcal{X}$D. For single-view 3D generation, a set of anchor views are first generated with single view condition, and then we generate more views by interpolating the camera poses of these anchor views. All the generated views are directly used to optimize a 3D-GS model for 2k iterations with photometric, LPIPS and mask loss functions . 4D generation requires the modeling of dynamic, so we use 4D Gaussian Splatting (4DGS) (Wu et al., 2024a) as backbone. Directly optimizing a 4D representation with 4D videos can be difficult to converge. Therefore, we first optimize a 3D-GS as 3D single-view generation manner for 500 iterations  and then optimize the dynamic deformation with the generated 4D video for 2k iterations with the same loss functions .

## I   FILTERING OBJAVERSE ANIMATION DATA

We perform filtering on the high-quality subset in Liang et al. (2024) (about 80K) to get the 44K dynamic data. Specifically, we evenly select keypoints on the first view of the object and then track the keypoints to get the object motion field as in Sec. 3. Based on the motion strength, we set a threshold (0.009) to filter data with low motion strength. The data filtering technique can not only remove data with small motion but also get the motion strength to be used in our multi-view temporal modules.

## J   LIMITATIONS

Gen$\mathcal{X}$D demonstrates remarkable performance in both 3D and 4D generation. However, there are two key limitations when applied to real-world scene generation. First, despite the abundance of 3D data, the diversity of real-world datasets is limited. For example, complex scenes (Zhou et al., 2018), typically feature simplistic camera trajectories, such as forward-facing views. In contrast, datasets focused on objects (Yu et al., 2023; Reizenstein et al., 2021), often have more varied camera paths, providing richer 3D information, but the object categories are generally few in number and the scene structure is quite simple. As a result, Gen$\mathcal{X}$D struggles with generating 360-degree views of complex scenes from single-view conditions.

Second, in 4D generation, temporal consistency and object motion are difficult to maintain during large camera movements in real-world scenarios. This limitation arises from the nature of available video data: video data typically provides limited object motion when the camera is moving quickly, whereas large object motion is often associated with static or slightly moving cameras. With such video data, our curated 4D data do not contain many samples with both large object and camera movement.

Both limitations are primarily due to the constraints of current datasets. However, with our proposed data curation pipeline and the increasing availability of public data, Gen$\mathcal{X}$D has the potential for significant improvement in future applications.

