# OpenReview forum: "GenXD: Generating Any 3D and 4D Scenes"
_ICLR.cc/2025/Conference — ICLR 2025 Poster_

### Official Review · Reviewer_oiM6 · 2024-10-31

**Soundness:** 3
**Presentation:** 3
**Contribution:** 4
**Rating:** 8
**Confidence:** 4

**Summary:**

In the paper the authors propose two main contributions: a curated dataset for 4D generation model learning, named CamVid-30K; and a model trained to generate in 3D or 4D given arbitrary condition images, named GenXD.

Authors proposed a detailed pipeline on how to combine existing techniques to curate a 4D scene datasets for model training. Including instance segmentation modules for static / dynamic decomposition, static structure-from-motion to recover camera parameters and sparse depth map, relative depth align with sparse depth map for spotting the dynamic object and introducing a motion strength factor as additional condition.

Authors proposed a new model GenXD to train on this dataset combining with other object-centric 3D/4D datasets and 3D scene datasets. They further design a $\alpha$-fusing strategy to better disentangle the spatial and temporal information in the data source. Experiments across various benchmark show impressive performance of the proposed method.

**Strengths:**

1. The data curation pipeline for transforming existing videos into trainable 4D dataset is quite useful, and the proposed curation pipeline and the CamVid-30K should be beneficial to the field.

2. Combining all source and sub-tasks' data (object/scene, 3D/4D) is fundamentally useful and a model trained on mixture of data should have better generalization ability. The proposed $\alpha$ parameter seems can be understood as an explicit control to switch between 3D and 4D generation given same conditions.

3. The results are promising and generally good. And extensive evaluations on multiple benchmarks show the effectiveness of the proposed method.

**Weaknesses:**

There are some minor errors or confusing points in the paper. I'll list some here and some in the following questions section.

1. In L253, "The keypoint $(u_i, v_i)^T$ in the $i$-th frame is first back-projected into world space to obtain the 3D keypoint $kp_i$". I agree here the $kp_i$ should be in world space, but according to Eq.(3) seems it's in the camera space? From my perspective the Eq.(3) is transforming image-space coordinates to camera-space coordinates, missing the step of transforming to world coordinates.

2. In all the figures with camera trajectory visualization, the legends and axis notations are very small and impossible to tell the actual information, also the trajectory only lies in a small region in the plot. I suggest authors remove the axis notations if they are too small, and zoom in to show the trajectory in a more detailed way.

3. In section 5.2 4D object Generation, it seems unfair to say "results in our method being $100\times$ faster", as the efficiency comes from using a different underlying 3D representation comparing to other methods, which are orthogonal to the proposed method. I think here using CLIP similarities for comparison is reasonable. Showing speed is fine but shouldn't be used as comparison.

4. I think in general the paper is with good results. But according to the task of the proposed method, I expect to see more scene level 3D or 4D generation results, including larger camera trajectories and failure examples.

**Questions:**

1. The statement in section 3 that "when the camera is moving while the object remains static, the motion strength is significantly smaller compared to videos with object motion" seems not so easy to understand. I assume the authors mean that this is the common case for natural captured video where cameras are still or moving in a slow motion?

2. Does the $\alpha$ needs to be explicitly set during training / inference? For example let network itself output the weight when dealing with 4D content and explicitly set it as 0 when dealing with 3D content. If so then it would be interesting to see given same conditions (more complicated than Figure 7) what would model outputs for different $\alpha$. Like given multi-frames of a static scene but telling model do 4D generation and given multi-step frames with dynamic objects and force model to do static 3D generation.

3. It's kind of confusing that the 5.4 ablation study is for model training or just inference after training? If it's after training, than the results in table 5 is somehow not so useful as it's trained with $\alpha$ but in inference time not allowed to use it, which would certainly lead to performance drop.

---

### Official Review · Reviewer_nZJ1 · 2024-11-01

**Soundness:** 3
**Presentation:** 3
**Contribution:** 3
**Rating:** 8
**Confidence:** 4

**Summary:**

1. This paper aims to jointly generate 3D and 4D objects and scenes with camera control.
2. This paper proposed multiview-temporal modules that disentangle camera and object movements and thus can learn from both 3D and 4D data. The proposed approach employs masked latent conditions to support a variety of conditioning views.
3. They construct a dataset CamVid-30K that consists of high-quality 4D data with camera poses for model training
4. Extensive experiments show that the proposed method can achieve comparable or better results than baselines in 3D/4D object/scene generation.

**Strengths:**

1. The proposed method is the first to generate any 3D and 4D scenes with camera control and an arbitrary number of condition frames.
2. The proposed multiview-temporal modules with alpha-fusing enable separate multi-view and temporal information and effectively conduct both 3D and 4D generation.
3. The paper constructs a new dataset for 4D scene generation. The dataset and the data curation pipeline potentially benefit the following video generation with camera control and 4D generation.
4. The paper is well-written and easy to follow.

**Weaknesses:**

**Experiments**:

1. In the experiment of 4D object generation, some relevant references and comparisons are missing, such as Consistent4D [1] and STAG4D [2].  Since these works are open-source, it would strengthen the paper to include these baselines or clarify why they are not suitable for comparison. They take single-view video as input, which should be applicable for this work.
2. In Table 3, it would also be beneficial to report temporal consistency metrics (e.g., FVD), as temporal consistency is critical for 4D object generation.

**Minor Points:**

1. Clarifying the selection process for the 44K dynamic data in Objaverse-XL would be helpful. According to Diffusion4D [Liang et al. (2024)], ~323K dynamic objects were collected. For instance, what filters were applied in this work? Will the selected dynamic objects be publicly available? Adding these details in the Appendix would enhance transparency.
2. Some technical details are missing: What is the maximum number of frames the model supports? Additionally, in Table 3, Zero-1-to-3 and RealFusion were originally designed for 3D reconstruction—how were they adapted for 4D generation in this work?

[1] Jiang, Yanqin, et al. "Consistent4d: Consistent 360 {\deg} dynamic object generation from monocular video." ICLR 2024.

[2] Zeng, Yifei, et al. "Stag4d: Spatial-temporal anchored generative 4d gaussians." ECCV 2024.

**Questions:**

1. In the top case of Figure 10, the results from the proposed method appear off-center, possibly due to an inappropriate object-to-image occupancy ratio in the input images. Adjusting this ratio might improve the alignment of the results.
2. If the learnable fusion weight, alpha, is set to 1, would it enable video generation based on the first frame? With alpha at 1, only the outputs from the temporal modules would contribute.

---

### Official Review · Reviewer_x1Cy · 2024-11-02

**Soundness:** 1
**Presentation:** 2
**Contribution:** 2
**Rating:** 3
**Confidence:** 5

**Summary:**

The paper trained a video generation model that can control camera trajectory and magnitude of motion and supports multiple frame conditioning.

**Strengths:**

The paper shares technical details on how to annotate magnitude motion and camera poses from in the wild videos. The alpha-fusion layers for motion disentangle seems an interesting design.

**Weaknesses:**

First, I feel the claim of being able to perform 4D generation is an over-claim to me. 4D generation requires the capability of either directly generating 4D representations such as dynamic 3D GS, or at least generating synchronized multi-view videos like in SV4D. Neither of these capabilities were presented in the main paper. In table 1, the capability of generating synchronized videos were not discussed, and to me, this is a severe misrepresentation. It would be more appropriate for the author to rebrand their method as a motion-controllable and 3D-aware video model.

2nd, although the idea of using alpha-fusion seems interesting, it is currently not properly evaluated. It did not show how changing alpha values affects the magnitude of generated motions, and it did not evaluate the camera control accuracy as other related papers did. Reporting CLIP-score and FID is not enough to reflect the accuracy of the proposed capability of the method.

3rd, a minor point, I am not sure promoting the capability of taking multiple image input can be regarded as a major technical contribution, given it is already supported in prior works including CAT3D, and it is conceptually trivial to be implemented in most video generation models.

**Questions:**

The author should provide rigorous analysis of the accuracy of the camera-controll capability, and how changing alpha values affects the generated motions.

---

### Official Review · Reviewer_rGgP · 2024-11-03

**Soundness:** 2
**Presentation:** 2
**Contribution:** 3
**Rating:** 6
**Confidence:** 5

**Summary:**

This paper proposes GENXD, a latent diffusion model for 3D/4D generation of objects or scenes. Specifically, it adopts masked latent conditions to support various number of input views, and the alpha-fusing mechanism allows joint training on 3D and 4D data. Considering the lack of 4D scene dataset, the authors further curated a new dataset, CAMVID-30K, by estimating camera with a SfM-based method and filtering out videos without object motion. Qualitative and quantitative results show that the proposed method generates comparable or slightly more satisfactory outputs than corresponding prior arts.

**Strengths:**

S1: Sensible model design
Although the masked latent conditioning is not new, the architectural modification upon SVD is sensible and allows joint training on 3D and 4D data.

S2: General model for 3D and 4D generation
The proposed model is capable of 3D and 4D generation of both object-centric and scene-level videos, which is more general than most prior methods. On a side note, the authors should also include MotionCtrl in Table 1.

S3: Good writing
The paper is well-written and easy to follow overall.

**Weaknesses:**

W1: Limitation of camera pose estimation
The proposed camera pose estimation relies on segmentation of all moving pixels. However, in scenarios where camera moves independently of object motion, especially when camera motion is large or objects take up a large portion of the scene, it would be challenging to estimate accurate camera pose. Does the method assume that these cases do not exist in the dataset?

W2: Quality of 3D object generation
The results of 3D object generation seem to be of comparable or worse quality than the prior state-of-the-arts both qualitatively (Figure 10) and quantitatively (Table 6). Moreover, the quantitative evaluation is incomplete since some more recent methods (Zero123XL, Magic123, SV3D, EscherNet, etc) are missing and the metric is limited to CLIP-I only, while prior works usually report metrics like LPIPS, SSIM, Chamfer Distance, 3D IoU (on 3D object datasets like Google Scanned Objects).

W3: Evaluation of 4D object generation
Again, the quantitative evaluation for 4D object generation is limited to the CLIP-I metric and more recent methods like STAG4D and DreamGaussian4D are missing. Also, it is unclear if the metrics in Table 3 are calculated on the training (synthesized) video frames only or on densely sampled views and timestamps. Since the proposed method optimizes 4D-GS only on one camera orbit without SDS loss, I suspect that the outputs look good on these frames/views but worse than other methods in novel views.

W4: Small camera motion in 4D scene generation
All the presented results on 4D scene generation seem to have smaller camera motion compared to results shown in prior work like MotionCtrl. Although the results in Figure 5 and supplemental video show decent temporal consistency and motion, I’m wondering if it is limited to camera trajectories without much deviation from the input view.

W5: Lack of results on motion strength control
While the paper emphasizes the contribution of motion strength control, there is only one example of a simple driving scene. It would be more insightful to show more diverse motion cases to understand the effeteness and limitations of it.

**Questions:**

Q1: Following W1, what are the assumption and failure cases of the proposed camera estimation?

Q2: Following W3, please describe how the metric is calculated in detail for fair comparison against prior methods.

---

### Meta-Review · Area_Chair_fUWH · 2024-12-24

**Metareview:**

This paper introduces an approach for sequential image generation that can properly depict 3D and 4D scenes. The key idea is to investigate camera and object movements jointly, which leads to curating real-world video to make a new CamVid-30k dataset. The proposed pipeline GenXD has new multiview-temporal modules, which can disentangle camera and object movements.

The strengths of this paper are summarized as follows:
- Sensible model design
- Interesting idea for motion disentangle
- First approach that proposes a general model for 3D and 4D generation
- New dataset for 4D scene reconstruction
- Good paper writing, easy to follow

The weaknesses of this paper are summarized as follows:
- Limitation of camera pose estimation - possibility of failure cases
- Missing references (Consistent4D and STAG4D)
- Minor questions regarding evaluation metrics and datasets

Overall, AC confirms that the experiment is extensive, and the results are compelling in the various cases. Although the paper received divergent scores, particularly the rejection score by x1Cy, AC confirms that the merits in this paper outweigh the concerns raised by x1Cy.

**Additional Comments On Reviewer Discussion:**

This paper received diverged scores {3, 6, 8, 8}. Overall, the reviewers weigh more on the value of the pioneering attempt for 3D and 4D scene generation. AC notes that the authors provide detailed and impressive feedback on the reviewers' questions. Specifically, the authors provide a clear summary that shows the merit of the proposed approach compared with other baseline approaches that can produce image-to-4D generation, video-to-4D generation, 4D scene generation, 3D object generation, and 3D scene generation. Moreover, the authors provide an anonymous webpage to provide additional results.

Regarding each reviewer's comment, the reviewer rGgp requested clarification on the camera pose estimation module, evaluation of the additional datasets, and evaluation with more 4D object generation tasks. The authors provide thorough evaluation results that compare the proposed approach with Zero123, Zero123-XL, EscherNet, Consistent4D, DreamGaussian4D, and STAG4D. The authors also offer a comparison with MotionCtrl and Camera Ctrl. The reviewer rGgp clarified that the additional results were convincing and increased the rating. The reviewer xZJ1 asks about missing references, evaluation metrics, data selection procedure, and missing technical details. The authors provide thorough feedback by providing additional comparisons. The reviewer rGgp was satisfied with the comments and highlighted that this paper is pioneering. The reviewer oiM6 provided a constructive review, mostly about clarification of the technical details and more results, and questions on the ablation study. The reviewer also mentioned that the authors' rebuttal clarified initial concerns and stated that this work is well-motivated and evaluation is sufficient.

In particular, the reviewer x1Cy provided a short initial review and asked about the misleading arguments, evaluation of alpha-fusion, and camera control accuracy. The reviewer also requested a comparison with SV4D and an unclear setting about 4D generation. AC sees that the authors provided a thorough rebuttal to this request. The reviwer x1Cy states that still the argument regarding 4D generation is still misleading and mention provided results are limited and may cherry-picked. During the Reviewer-AC discussion phase, AC requests the reviewer's opinion x1Cy again whether the rebuttal and other reviewers' comments would change the score. However, the reviewer x1Cy did not reply.

---

### Decision · Program_Chairs · 2025-01-22

Accept (Poster)